# DNA barcoding for the assessment of marine and coastal fish diversity from the Coast of Mozambique

Valdemiro Muhala[1,2]*, Aurycéia Guimarães-Costa[1]*, Isadola Eusébio Macate[1,3], Luan Pinto Rabelo[1], Adam Rick Bessa-Silva[1], Luciana Watanabe[1], Gisele Damasceno dos Santos[1], Luísa Sambora[4], Marcelo Vallinoto[1], Iracilda Sampaio[1]

**1** Laboratório de Evolução, Universidade Federal do Pará, Alameda Leandro Ribeiro, Aldeia, Bragança, Pará, Brazil, **2** Divisão de Agricultura, Instituto Superior Politécnico de Gaza, Chókwè, Mozambique, **3** Departamento de Ciências Agrárias e Ambientais, Universidade Estadual de Santa Cruz, Ilheus, BA, Brazil, **4** Departamento de Produção Agrária, Escola Superior de Desenvolvimento Rural, Universidade Eduardo Mondlane, Vilankulos, Moçambique

* valdemiro.muhala@ispg.ac.mz (VM); auryceia@yahoo.com.br (AG-C)

## Abstract

The ichthyological provinces of Mozambique are understudied hotspots of global fish diversity. In this study, we applied DNA barcoding to identify the composition of the fish fauna from the coast of Mozambique. A total of 143 species belonging to 104 genera, 59 families, and 30 orders were identified. The overall K2P distance of the COI sequences within species ranged from 0.00% to 1.51%, while interspecific distances ranged from 3.64% to 24.49%. Moreover, the study revealed 15 threatened species according to the IUCN Red List of Threatened Species, with elasmobranchs being the most represented group. Additionally, the study also uncovered four new species that were not previously recorded in this geographic area, including *Boleophthalmus dussumieri*, *Maculabatis gerrardi*, *Hippocampus kelloggi*, and *Lethrinus miniatus*. This study represents the first instance of utilizing molecular references to explore the fish fauna along the Mozambican coast. Our results indicate that DNA barcoding is a dependable technique for the identification and delineation of fish species in the waters of Mozambique. The DNA barcoding library established in this research will be an invaluable asset for advancing the understanding of fish diversity and guiding future conservation initiatives.

## 1. Introduction

The Mozambique Channel comprises an arm of the Indian Ocean located between the Southeast African countries of Madagascar and Mozambique, stretching 1,600 km along the coast [1]. It is recognized as an important diversity hotspot due to a variety of coastal ecosystems that distinguish it from other Western Indian Ocean (WIO) biogeographic provinces [2, 3]. These systems provide a range of habitats for both animals and plants, making the region abundant in aquatic species [4–6]. The diverse habitats include a large rocky coastline in the

**Funding:** This research was financed by Conselho Nacional de Desenvolvimento Científico e Tecnológico (CNPq) through a research project 407536/2021-3, 309916/2021-6 and the APC was funded by Pro-Reitoria de Pesquisa e Pós-Graduação of the Universidade Federal do Pará. The funders had no role in the study design, data collection and analysis, decision to publish, or preparation of the manuscript.

**Competing interests:** The authors have declared that no competing interests exist.

north with extensive coral reefs, a wider platform in the central part with river outflows and weaker ocean currents that form sandy banks, estuaries, and deltas, and a southern coast characterized by wide beaches, bays, and seagrass beds with numerous endemic species [7–9].

Investigations into fish species diversity on the west coast of Africa remain in their early stages, largely due to a scarcity of both traditional taxonomists and those employing molecular tools for taxonomic inference within various fish groups [10]. Accurate classification and identification of fish species are important not only for taxonomists but also for various types of research, including fisheries, natural resource surveys, forensic studies, the discovery of cryptic species, and the identification of species not previously known and their conservation status within a specific biogeographic region [11–15]. The impacts of lacking knowledge on ichthyofauna may have even greater repercussions because research focused on fishing activity can be seriously compromised [16].

Historically, surveys of fish fauna off the coast of Mozambique have primarily relied on traditional methods of identification, which use morphological features and meristic counts to distinguish and define fish species [17, 18]. However, due to subtle morphological differences, complex evolutionary patterns in certain groups, and the existence of cryptic species, these methodologies may not always overcome the challenges of accurately identifying species, resulting in an underestimation of the true diversity of ichthyofauna [19–21].

These concerns are particularly significant for international organizations focused on biodiversity conservation, which have traditionally relied on the rigid concept of species. This concept has recently been widely challenged by the more comprehensive and integrated approaches adopted by modern taxonomy for classifying living organisms. [22–24]. Although there is consensus that species represent the smallest independent evolutionary unit, their recognition remains a subject of debate [25, 26], potentially impacting conservation efforts. Furthermore, morphological identification of fish can prove challenging task if not conducted by specialists, leading to inaccurate identifications [27, 28]. Additionally, previous genetic studies have shown that molecular identification may not align with identifications based on morphological characteristics in some instances [29, 30]. This highlights the importance of carrying out a systematic and comprehensive inventory of ichthyofauna using standardized DNA-based methods.

In response to the limitations of classical taxonomy, DNA barcoding has emerged as a widely adopted research method for accurately identifying species [31–33]. This technique, which relies on the diversity of the Cytochrome C Oxidase I (COI) gene region within mitochondrial DNA, serves as an alternative to traditional taxonomic approaches. DNA barcoding has proven crucial in enhancing the accuracy of species identification and is especially valuable for differentiating between diverse and poorly understood flora and fauna that necessitate species delimitation [31–37].

Large-scale investigations of ichthyofaunas across various ecosystems have demonstrated that utilizing the cytochrome oxidase I gene captures a significant proportion of known diversity [39–42]. However, it has also helped to reveal previously unknown diversity [43–46]. This method is successful due to the availability of reliable DNA barcode reference libraries in the BOLD Systems (Barcode of Life Data Systems) and GenBank [38]. New DNA barcodes can be analysed with available data to check for potential taxonomic conflicts and improve taxonomic resolution [46].

Considering the remarkable potential of DNA barcoding to identify fish species and considering the still unknown fish assemblage in the coast of Mozambique, the aim of this study is to employ this approach to establish a DNA barcoding reference database for the composition of the fish fauna off the coast of Mozambique. This will have significant implications for our

understanding of diversity and could contribute to future conservation efforts for marine species in Mozambique.

## 2. Material and methods

### 2.1 Ethical statement

The samples analyzed in this study were obtained in accordance with the requirements of environmental legislation, approved by the Ministry of Fisheries (license no. 178MP11227B/20) and the Instituto Superior Politécnico de Gaza—ISPG (license no. 409/GDG/ISPG/090), for the collection and transportation of samples. The authorization also includes the use of anesthetic (5% Lidocaine) on the skin to minimize animal suffering, following the recommendations of the Herpetological Animal Care and Use Committee (HACC) of the American Society of Ichthyologists and Herpetologists.

### 2.2 Sampling area and specimen collection

The Mozambican coast is in Eastern Africa (Fig 1), spanning over 2.700 km of coastline along the Indian Ocean, second largest African coast [47, 48]. The coastal region showcases a range of diverse landscapes, including sandy beaches, dunes, forests, mangrove swamps, seagrass, and coral reefs. Three distinct ecological zones can be found along the coast, including the

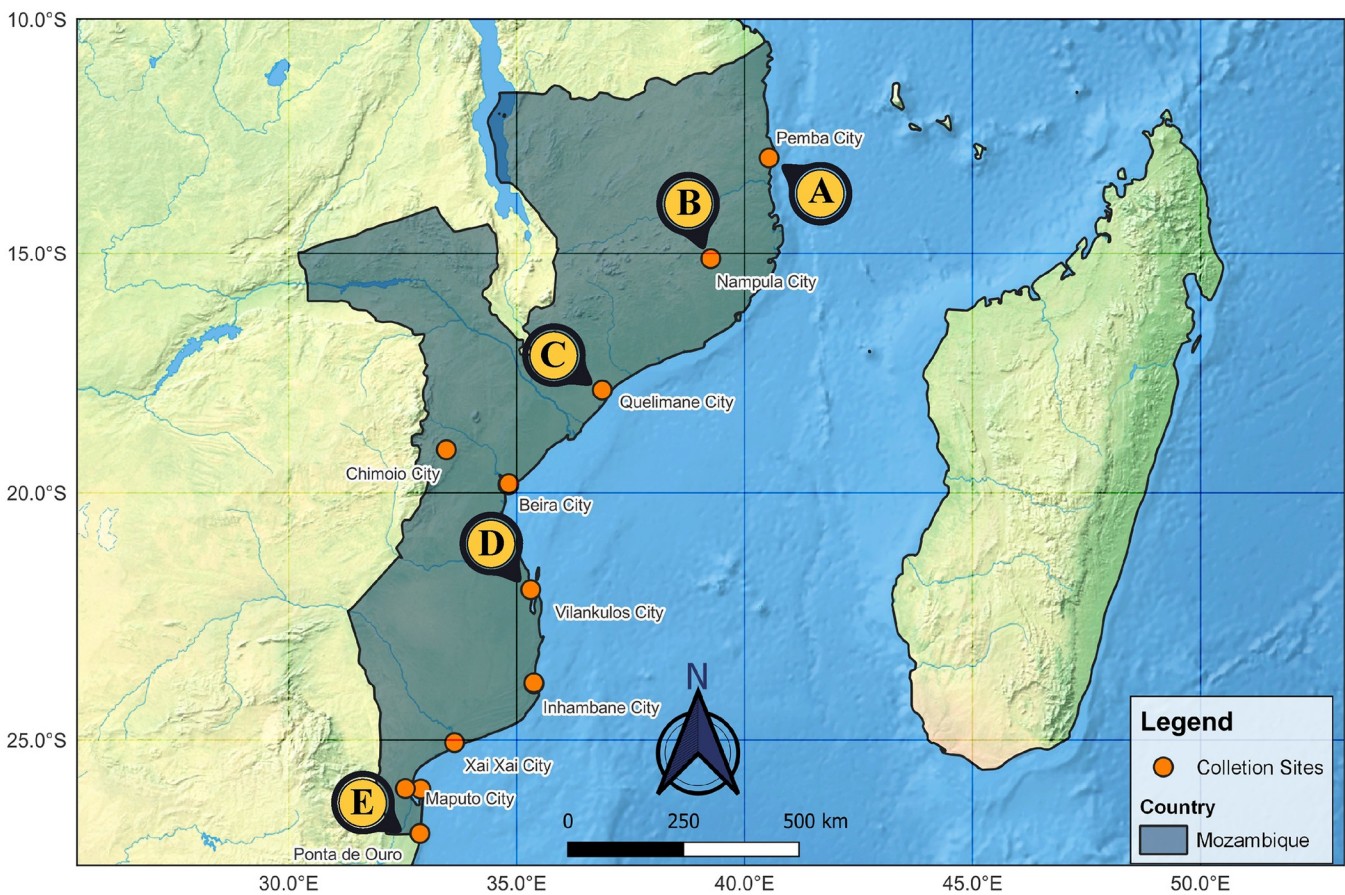

**Fig 1. Map of sample origin areas.** All the yellow dots correspond to the total areas collected along the coast. The points marked with A to E Are Specific illustrations of the different sampling areas.

Delagoa Basin in the south, the highly productive and rich Sofala Bank in the central region, and the São Lazaro Bank further north [48, 49].

Sampling was conducted randomly across all coastal provinces of the country from December 2019 to April 2022, considering both dry and rainy seasons (Fig 1). Different fishing gear (trawls, surface, and bottom gillnets) commonly used in the country's artisanal fisheries were used for specimen collection. The collected samples were stored in ice boxes and transported to the sorting site, where a substantial portion of the specimens was photographed, and tissues were removed. The specimens were identified to the lowest taxonomic level based on the nomenclature used by the vendors and using species identification keys specific to South-eastern Africa [50, 51].

Approximately 50 mg of muscle tissue was collected from each specimen and preserved in 1.5 ml Eppendorf-type microtubes filled with 96% ethanol. The samples were stored in a freezer set at -20˚C. After the initial collection of tissues, the specimens were transported to the Evolution Laboratory at the Institute of Coastal Studies of the Federal University of Para, located in Bragança, Pará, Brazil, for the purpose of generating genetic data.

## 2.3 Genomic DNA extraction, PCR amplification, and DNA sequencing

The extraction of DNA was carried out using the Wizard Genomic DNA Purification Kit (Promega Corporation, Madison, WI, USA) according to the manufacturer's protocol. Total genomic DNA was extracted from muscle or fin tissue of fish specimens. To analyse intra- and inter-variability, a partial COI gene from mitochondrial DNA was used as the molecular marker. The 5′ end of the COI gene was amplified using two pairs of universal primers for fish, FishF1/FishR1 and FishF2/FishR2 [41], resulting in a maximum of 622 bp fragment. The 5′ region of the COI gene was amplified by Polymerase Chain Reaction (PCR) in a 25 μL reaction mixture that consisted of 4 μL of dNTPs (1.25 mM), 2.5 μL of 10X buffer solution, 1 μL of $MgCl_2$ (25 mM), 0.25 μL of each primer (200 ng/μL), 1–1.5 μL of genomic DNA (100 ng/μL), 1 U of Taq DNA polymerase (5 U/μL), and purified water. The amplification process was as follows: an initial denaturation step at 94˚C for 3 minutes, followed by 35 cycles of denaturation at 95˚C for 1 minute, annealing at 50˚C-58˚C for 45 seconds, and extension at 72˚C for 45 seconds. The process was concluded with a final extension at 72˚C for 5 minutes. The positive reactions were then sequenced using an ABI 3500 automatic sequencer (Applied Biosystems).

## 2.4 Molecular data analysis

The final sequences were aligned using ClustalW [52], implemented in GENEIOUS 9.0.5 (https://www.geneious.com), and then translated into amino acids to check for potential stop codons using the MEGA 11 software. All the generated sequences were submitted to BOLD database. The high-quality sequences were compared using the National Canter for Biotechnology Information (NCBI) BLAST search engine and the Barcode of Life Data Systems (BOLD) database. The Eschmeyer's Catalog of Fishes repositor was used to compile the taxonomic information of the species, and the classifications of the families and orders followed the study by [53]. After analysing the sequences, parameters such as the presence of barcode gap, sequence length, GC content, and intra/inter genetic distances within and between families, genera, species, and minimum distances to the nearest neighbour were calculated using tools in the BOLD Systems platform using Kimura's two-parameter model and bootstrap of 1000 replicates [54]. Phylogenetic trees were constructed in MEGA 11 using the NJ method based on the Kimura 2-Parameter model with 1000 bootstrap pseudoreplicates [54, 55] and the online tool Interactive Tree of Life (iTOL) [56] were used to visualize and edit the tree.

Analysis of Barcode Index Numbers (BIN) [57] was conducted for species delimitation, based on uncorrected p-distances, which provide a single BIN for each Operational Taxonomic Unit (OTU) obtained from the COI sequences. The analysis of BINs reveals the maximum intra-specific distance and the minimum inter-specific distance that overlap and allow for species identification. The aligned sequences, primer pairs, trace files, taxonomic information, and collection data were deposited in the Barcode of Life Data Systems under the project MOZFH [57]. All Barcode Index Numbers (BIN) and GenBank accession number are included in the Table 1.

## 3. Results

### 3.1 DNA barcoding results

In this study, we generated a final alignment with 419 sequences of the barcode region of COI gene with a length of 622 bp. Of these, we only used sequences with a length of 400 up to 622 bp for genetic analysis. The data correspond to 143 species, 104 genera, 59 families, 30 orders, and 4 unidentified taxa (Table 1 and S1 Table). All samples were assessed morphologically and then subjected to DNA barcoding evaluation to confirm their identification. There were two classes (Teleost and Elasmobranch) of fish samples, including 392 and 27 respectively, which were acquired from local artisanal fishermen at landing sites and fish markets. For more details of the NJ tree clustering and composition see (S1 Fig).

The species composition was represented by Carangidae (16 species), Sparidae (10 species), Serranidae (9 species), Haemulidae and Lutjanidae (7 species each), Scombridae and Mugilidae (6 species each). The remaining families were composed of fewer than 5 species (Table 1). The sequences generated in this study were all submitted to the BOLD Systems and the species list details are provided in (S1 Table).

The final analysed database had no insertion, deletion, or stop codon, indicating that it accurately represents the mitochondrial COI fragment. Sequences with poor quality DNA were excluded from the analysis. The nucleotide content showed the following average frequency: G = 17.77%, C = 28.74%, A = 23.88% and T = 29.61% with an average GC content of 46.51%.

The NJ tree included 143 distinct OTUs with high bootstrap support of 70/99 (Fig 2). The specimens belonging to the same species exhibited clustering based on both morphological

**Table 1. List of the 143 fish species, from Mozambican coast waters, which were DNA barcoded.**

| Order | Family | Species | Authorship | Status Conservation (IUCN) | BIN | Accession |
|---|---|---|---|---|---|---|
| Carcharhiniformes | Carcharhinidae | *Carcharhinus leucas* | (Müller & Henle, 1839) | VU | BOLD:AAA6060 | OR284674 |
| | | *Loxodon macrorhinus* | (Müller & Henle, 1839) | NT | BOLD:AAA6751 | OR284672 |
| | Sphyrnidae | *Sphyrna lewini* | (Griffith & Smith, 1834) | CR | BOLD:AAA2402 | OR284675 |
| Myliobatiformes | Dasyatidae | *Himantura uarnak* | (Gmelin, 1789) | EN | BOLD:AAB7830 | OR284662 |
| | | *Maculabatis gerrardi* | (Gray, 1851) | EN | BOLD:AAA8673 | OR284658 |
| | | *Pastinachus ater* | (Macleay, 1883) | VU | BOLD:AAC1400 | OR284664 |
| | Myliobatidae | *Rhinoptera* sp. | (Cuvier, 1829) | - | BOLD:AAC7667 | OR284652 |
| | Plesiobatidae | *Plesiobatis daviesi* | (Wallace, 1967) | LC | BOLD:AAF2578 | OR284653 |
| Acanthuriformes | Acanthuridae | *Acanthurus mata* | (Cuvier, 1829) | LC | BOLD:AAE4025 | OR284615 |
| | | *Naso brevirostris* | (Cuvier, 1829) | LC | BOLD:AAC1635 | OR284418 |
| Anguilliformes | Muraenesocidae | *Muraenesox bagio* | (Hamilton, 1822) | LC | BOLD:ACK7558 | OR284694 |
| Beloniformes | Hemiramphidae | *Hemiramphus far* | (Forsskål, 1775) | - | BOLD:AAC0565 | OR284496 |

*(Continued)*

**Table 1.** (Continued)

| Order | Family | Species | Authorship | Status Conservation (IUCN) | BIN | Accession |
|---|---|---|---|---|---|---|
| Carangiformes | Carangidae | *Alepes djedaba* | (Forsskål, 1775) | LC | BOLD:AAB5772 | OR284371 |
| | | *Caranx ignobilis* | (Forsskål, 1775) | LC | BOLD:AAB0587 | OR284379 |
| | | *Caranx papuensis* | (Alleyne & Macleay, 1877) | LC | BOLD:ACF4541 | OR284388 |
| | | *Caranx sexfasciatus* | (Quoy & Gaimard, 1825) | LC | BOLD:AAB0584 | OR284383 |
| | | *Decapterus macarellus* | (Cuvier, 1833) | LC | BOLD:AAB6796 | OR284404 |
| | | *Gnathanodon speciosus* | (Forsskål, 1775) | LC | BOLD:AAB7462 | OR284373 |
| | | *Megalaspis cordyla* | (Linnaeus, 1758) | LC | BOLD:AAB5271 | OR284374 |
| | | *Parastromateus niger* | (Bloch, 1795) | LC | BOLD:AAB3884 | OR284400 |
| | | *Platycaranx chrysophrys* | (Kimura, Takeuchi & Yadome, 2022) | - | BOLD:AAB2977 | OR284393 |
| | | *Platycaranx malabaricus* | (Kimura, Takeuchi & Yadome, 2022) | - | BOLD:AAB3474 | OR284391 |
| | | *Scomberoides commersonnianus* | (Lacepède, 1801) | LC | BOLD:AAB6417 | OR284447 |
| | | *Scomberoides tol* | (Cuvier, 1832) | LC | BOLD:AAB6418 | OR284444 |
| | | *Scyris indica* | (Cuvier & Valenciennes, 1833) | - | BOLD:AAB7826 | OR272032 |
| | | *Selar crumenophthalmus* | (Bloch, 1793) | LC | BOLD:AAB0870 | OR284408 |
| | | *Turrum coeruleopinnatum* | (Whitley, 1932) | - | BOLD:AAD2297 | OR284399 |
| | | *Turrum fulvoguttatum* | (Whitley, 1932) | - | BOLD:AAC2745 | OR284396 |
| | Coryphaenidae | *Coryphaena hippurus* | (Linnaeus, 1758) | LC | BOLD:AAA5277 | OR284696 |
| | Rachycentridae | *Rachycentron canadum* | (Linnaeus, 1766) | LC | BOLD:AAB2939 | OR284680 |
| Centrarchiformes | Terapontidae | *Terapon jarbua* | (Forsskål, 1775) | LC | BOLD:AAA9351 | OR284426 |
| Cichliformes | Cichlidae | *Oreochromis mossambicus* | (Peters, 1852) | VU | BOLD:AAA8511 | OR284549 |
| | | *Oreochromis niloticus* | (Linnaeus, 1758) | LC | BOLD:ADI0792 | OR284554 |
| | | *Oreochromis aureus* | (Steindachner, 1864) | - | BOLD:AAC9904 | OR284556 |
| | | *Tilapia* sp. | (Smith, 1840) | - | BOLD:ABZ6465 | OR284560 |
| Chaetodontiformes | Leiognathidae | *Leiognathus equulus* | (Forsskål, 1775) | LC | BOLD:AAB2487 | OR284620 |
| Clupeiformes | Chirocentridae | *Chirocentrus dorab* | (Forsskål, 1775) | LC | BOLD:AAC2273 | OR284640 |
| | Clupeidae | *Hilsa kelee* | (Cuvier, 1829) | LC | BOLD:AAC0856 | OR284533 |
| | | *Sardinella gibbosa* | (Bleeker, 1849) | LC | BOLD:AAB7263 | OR284538 |
| | | *Sardinella jussieu* | (Lacepède, 1803) | DD | BOLD:AEW0759 | OR284540 |
| | Pristigasteridae | *Pellona ditchela* | (Valenciennes, 1847) | LC | BOLD:AAD4543 | OR284631 |
| | Engraulidae | *Thryssa vitrirostris* | (Gilchrist & Thompson, 1908) | LC | BOLD:AAE3536 | OR284635 |
| Cypriniformes | Cyprinidae | *Hypophthalmichthys molitrix* | (Valenciennes, 1844) | NT | BOLD:AAF6633 | OR284617 |
| Ephippiformes | Ephippidae | *Platax teira* | (Forsskål, 1775) | LC | BOLD:AAC5812 | OR284532 |
| | | *Tripterodon orbis* | (Playfair, 1867) | - | BOLD:AAD7693 | OR284531 |
| Gadiformes | Merlucciidae | *Merluccius paradoxus* | (Franca, 1960) | - | BOLD:AAC4936 | OR284610 |
| Gerreiformes | Gerreidae | *Gerres longirostris* | (Lacepède, 1801) | LC | BOLD:AAE6359 | OR284518 |
| | | *Gerres oyena* | (Forsskål, 1775) | LC | BOLD:AAC1291 | OR284515 |
| | | *Gerres methueni* | (Regan, 1920) | - | BOLD:AAF8786 | OR284523 |
| Gobiiformes | Gobiidae | *Boleophthalmus dussumieri* | (Valenciennes, 1837) | LC | BOLD:ABZ8637 | OR284562 |
| | | *Periophthalmus argentilineatus* | (Valenciennes, 1837) | LC | BOLD:AAF8789 | OR284567 |
| Istiophoriformes | Istiophoridae | *Istiompax indica* | (Cuvier, 1832) | DD | BOLD:ACF5077 | OR284572 |
| Labriformes | Labridae | *Cheilinus trilobatus* | (Lacepède, 1801) | LC | BOLD:AAB4188 | OR284596 |
| | | *Cheilio inermis* | (Forsskål, 1775) | LC | BOLD:AAA6101 | OR284546 |
| Lobotiformes | Lobotidae | *Lobotes surinamensis* | (Bloch, 1790) | LC | BOLD:AAC1878 | OR284651 |

(*Continued*)

**Table 1.** (Continued)

| Order | Family | Species | Authorship | Status Conservation (IUCN) | BIN | Accession |
|---|---|---|---|---|---|---|
| Lutjaniformes | Haemulidae | *Diagramma picta* | (Thunberg, 1792) | - | BOLD:AAD4477 | OR284508 |
| | | *Plectorhinchus gaterinus* | (Forsskål, 1775) | LC | BOLD:AAH9156 | OR284504 |
| | | *Plectorhinchus gibbosus* | (Lacepède, 1802) | LC | BOLD:ACK7983 | OR284512 |
| | | *Plectorhinchus flavomaculatus* | (Cuvier, 1830) | - | BOLD:AAC4020 | OR284499 |
| | | *Pomadasys kaakan* | (Cuvier, 1830) | LC | BOLD:AAD6593 | OR284369 |
| | | *Pomadasys maculatus* | (Bloch, 1793) | LC | BOLD:AAB3687 | OR284367 |
| | Lutjanidae | *Aprion virescens* | (Valenciennes, 1830) | LC | BOLD:AAB8692 | OR284415 |
| | | *Lutjanus argentimaculatus* | (Forsskål, 1775) | LC | BOLD:AAB2440 | OR284348 |
| | | *Lutjanus bengalensis* | (Bloch, 1790) | LC | BOLD:AAB7901 | OR284352 |
| | | *Lutjanus erythropterus* | (Bloch, 1790) | LC | BOLD:AAB3276 | OR284351 |
| | | *Lutjanus fulviflamma* | (Forsskål, 1775) | LC | BOLD:ADF5681 | OR284341 |
| | | *Lutjanus lutjanus* | (Bloch, 1790) | LC | BOLD:AAA8168 | OR284345 |
| | | *Pristipomoides filamentosus* | (Valenciennes, 1830) | LC | BOLD:AAB4096 | OR284363 |
| Mugiliformes | Mugilidae | *Chelon dumerili* | (Steindachner, 1870) | DD | BOLD:AAI6001 | OR284442 |
| | | *Crenimugil buchanani* | (Bleeker, 1853) | - | BOLD:AAE3561 | OR284437 |
| | | *Moolgarda* sp1 | (Moolgarda) | - | BOLD:AAC4146 | OR284440 |
| | | *Moolgarda* sp2 | (Moolgarda) | - | BOLD:AAG6597 | - |
| | | *Planiliza melinoptera* | (Valenciennes, 1836) | - | BOLD:AAD7507 | - |
| | | *Planiliza macrolepis* | (Smith, 1846) | LC | BOLD:AAC0698 | OR284431 |
| Pempheriformes | Pempheridae | *Pempheris connelli* | (Randall & Victor, 2015) | - | BOLD:AAC6084 | OR284691 |
| Perciformes | Caesionidae | *Caesio caerulaurea* | (Lacepède, 1801) | LC | BOLD:AAB4823 | OR284336 |
| | | *Caesio xanthonota* | (Bleeker, 1853) | LC | BOLD:AAE8330 | OR284339 |
| | | *Caesio* sp. | (Lacepède, 1801) | - | BOLD:AAB4822 | OR284335 |
| | | *Pterocaesio tile* | (Cuvier, 1830) | LC | BOLD:AAB4821 | OR284331 |
| | Dinopercidae | *Dinoperca petersi* | (Day, 1875) | - | BOLD:AAD0006 | OR284361 |
| | Menidae | *Mene maculata* | (Bloch & Schneider, 1801) | - | BOLD:AAB4862 | OR284410 |
| | Polynemidae | *Polydactylus malagasyensis* | (Motomura & Iwatsuki, 2001) | - | BOLD:AAB7311 | OR284649 |
| | Scaridae | *Chlorurus atrilunula* | (Randall & Bruce, 1983) | LC | BOLD:AAE8961 | OR284527 |
| | | *Chlorurus sordidus* | (Forsskål, 1775) | LC | BOLD:AAB6670 | OR284526 |
| | | *Scarus psittacus* | (Forsskål, 1775) | LC | BOLD:AAB8901 | OR284525 |
| | | *Scarus quoyi* | (Valenciennes, 1840) | LC | BOLD:AAD0849 | OR284524 |
| | Serranidae | *Variola louti* | (Forsskål, 1775) | LC | BOLD:AAC5719 | OR284591 |
| | | *Cephalopholis argus* | (Schneider, 1801) | LC | BOLD:AAC4474 | OR284493 |
| | | *Cephalopholis boenak* | (Bloch, 1790) | LC | BOLD:AAB3684 | OR284494 |
| | | *Epinephelus rivulatus* | (Valenciennes, 1830) | LC | BOLD:ACZ9919 | OR284480 |
| | | *Epinephelus coioides* | (Hamilton, 1822) | LC | BOLD:AAB8391 | OR284486 |
| | | *Epinephelus areolatus* | (Forsskål, 1775) | LC | BOLD:AAA9822 | OR284490 |
| | | *Epinephelus macrospilos* | (Bloch, 1793) | - | BOLD:AAE1882 | OR284479 |
| | | *Epinephelus malabaricus* | (Bloch & Schneider, 1801) | LC | BOLD:AAB8389 | OR284481 |
| | | *Epinephelus merra* | (Bloch, 1793) | LC | BOLD:AAB8387 | OR284492 |
| | Siganidae | *Siganus stellatus* | (Forsskål, 1775) | LC | BOLD:AAB2341 | OR284356 |
| | | *Siganus sutor* | (Valenciennes, 1835) | LC | BOLD:AAB6556 | OR284353 |
| | Sillaginidae | *Sillago sihama* | (Forsskål, 1775) | LC | BOLD:AAA7600 | OR284541 |
| | Platycephalidae | *Cociella heemstrai* | (Knapp, 1996) | LC | BOLD:AAD7871 | OR284627 |

*(Continued)*

**Table 1.** (Continued)

| Order | Family | Species | Authorship | Status Conservation (IUCN) | BIN | Accession |
|---|---|---|---|---|---|---|
| Pleuronectiformes | Citharidae | *Citharoides macrolepis* | (Gilchrist, 1904) | DD | BOLD:AAB1336 | OR284684 |
| | Cynoglossidae | *Cynoglossus attenuatus* | (Gilchrist, 1904) | LC | - | OR272033 |
| | Paralichthyidae | *Pseudorhombus arsius* | (Hamilton, 1822) | LC | BOLD:ACX7474 | OR284688 |
| | Psettodidae | *Psettodes erumei* | (Bloch & Schneider, 1801) | DD | BOLD:AAB6707 | OR284573 |
| Priacanthiformes | Priacanthidae | *Priacanthus hamrur* | (Forsskål, 1775) | LC | BOLD:AAB1643 | OR284626 |
| | | *Priacanthus arenatus* | (Cuvier, 1829) | LC | BOLD:AEV2303 | OR284623 |
| Rhinopristiformes | Rhinidae | *Rhynchobatus djiddensis* | (Forsskål, 1775) | CR | BOLD:AAC4065 | OR284670 |
| | | *Rhynchobatus australiae* | (Whitley, 1939) | CR | BOLD:AAC4063 | OR284668 |
| | Rhinobatidae | *Acroteriobatus leucospilus* | (Giltay, 1928) | - | BOLD:AAI0445 | OR284671 |
| Scombriformes | Nomeidae | *Cubiceps whiteleggii* | (Waite, 1894) | - | BOLD:AAB5183 | OR270958 |
| | Scombridae | *Acanthocybium solandri* | (Cuvier, 1832) | LC | BOLD:AEZ9639 | OR270954 |
| | | *Euthynnus affinis* | (Cantor, 1849) | LC | BOLD:AAB2798 | OR270940 |
| | | *Katsuwonus pelamis* | (Linnaeus, 1758) | LC | BOLD:AAA5421 | OR270939 |
| | | *Rastrelliger kanagurta* | (Cuvier, 1816) | DD | BOLD:AAA9666 | OR284326 |
| | | *Scomberomorus plurilineatus* | (Fourmanoir, 1966) | DD | BOLD:AAE9334 | OR270952 |
| | | *Scomberomorus commerson* | (Lacepède, 1800) | NT | BOLD:AAB5513 | OR270947 |
| | | *Thunnus albacares* | (Bonnaterre, 1788) | LC | BOLD:AAA7352 | OR270942 |
| | Trichiuridae | *Trichiurus* sp. | (Linnaeus, 1758) | - | BOLD:AAB0164 | OR284528 |
| | Ariommatidae | *Ariomma indicum* | (Day, 1871) | - | BOLD:AAB5940 | OR270959 |
| Scorpaeniformes | Platycephalidae | *Platycephalus indicus* | (Linnaeus, 1758) | DD | BOLD:AAB2371 | OR284576 |
| Siluriformes | Pangasiidae | *Pangasius djambal* | (Bleeker, 1846) | LC | BOLD:AAE3237 | OR284586 |
| | Ariidae | *Plicofollis polystaphylodon* | (Bleeker, 1846) | - | BOLD:AEW1876 | OR284581 |
| Spariformes | Lethrinidae | *Lethrinus nebulosus* | (Forsskål, 1775) | LC | BOLD:ABY6363 | OR284420 |
| | | *Lethrinus miniatus* | (Forster, 1801) | LC | BOLD:AAC8078 | OR284423 |
| | | *Lethrinus borbonicus* | (Valenciennes, 1830) | LC | BOLD:AAB0511 | OR284419 |
| | Sparidae | *Acanthopagrus vagus* | (Peters, 1852) | VU | BOLD:AAD8720 | OR284460 |
| | | *Acanthopagrus berda* | (Forsskål, 1775) | LC | BOLD:AAC4120 | OR284456 |
| | | *Chrysoblephus puniceus* | (Gilchrist & Thompson, 1908) | LC | BOLD:AAD8479 | OR284471 |
| | | *Crenidens crenidens* | (Forsskål, 1775) | LC | BOLD:AAE4408 | OR284468 |
| | | *Dentex macrophthalmus* | (Bloch, 1791) | LC | BOLD:AAL1497 | OR284477 |
| | | *Lithognathus mormyrus* | (Linnaeus, 1758) | LC | BOLD:AAD3990 | OR284465 |
| | | *Pagellus natalensis* | (Steindachner, 1903) | LC | BOLD:AAF8829 | OR284474 |
| | | *Rhabdosargus thorpei* | (Smith, 1979) | LC | BOLD:AAE3481 | OR284449 |
| | | *Rhabdosargus sarba* | (Forsskål, 1775) | LC | BOLD:AAB4844 | OR284454 |
| | Nemipteridae | *Scolopsis bimaculata* | (Rüppell, 1828) | LC | BOLD:AAD6249 | OR284619 |
| Syngnathiformes | Mullidae | *Parupeneus indicus* | (Shaw, 1803) | LC | BOLD:AAB0334 | OR284513 |
| | | *Upeneus vittatus* | (Forsskål, 1775) | LC | BOLD:ABZ7416 | OR284514 |
| | Fistulariidae | *Fistularia petimba* | (Lacepède, 1803) | LC | BOLD:AAB8439 | OR284590 |
| | Syngnathidae | *Hippocampus kelloggi* | (Jordan & Snyder, 1901) | VU | BOLD:AAF0667 | OR284677 |
| *Incertae sedis* | Sciaenidae | *Argyrosomus japonicus* | (Temminck & Schlegel, 1843) | EN | BOLD:AAC2818 | OR284602 |
| | | *Argyrosomus thorpei* | (Smith, 1977) | EN | BOLD:AAD8716 | OR284597 |
| | | *Otolithes ruber* | (Bloch & Schneider, 1801) | LC | BOLD:AAB6481 | OR284605 |
| | Sphyraenidae | *Sphyraena jello* | (Cuvier, 1829) | - | BOLD:AAA6102 | OR284644 |
| | | *Sphyraena chrysotaenia* | (Klunzinger, 1884) | - | BOLD:AAD0400 | OR284690 |

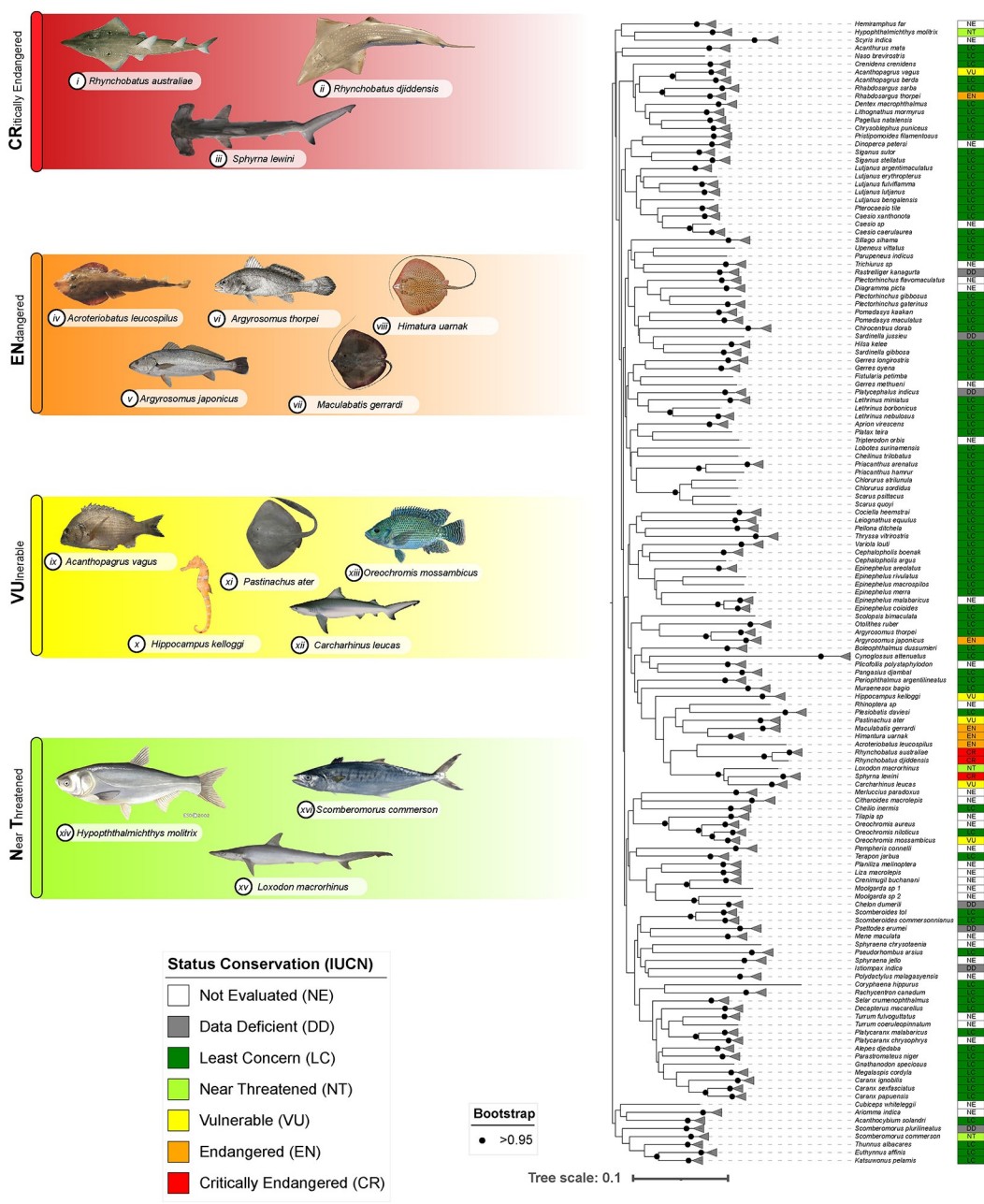

**Fig 2. NJ tree illustrating the relationships among Mozambican fish species based on DNA barcode sequences.** The NJ tree includes both teleost and elasmobranch fish species, constructed using K2P distances from 400 pb barcode sequences (COI). Bootstrap support values are represented by dots at the nodes. Images credits are given to i-iii, vii, xii—(Pollerspöck, J. & Straube, N. 2022, www.shark-references.com; iv - https://link.springer.com/article/10.1007/s12526-021-01208-6; v—Infofish Austrália; vi–FishBase; viii-https://indiabiodiversity.org/; ix-FishBase; xi - https://fishesofaustralia.net.au; xiii - https://www.fishesoftexas.org/; xiv -https://animaldiversity.org/; xv - https://fishider.org; xvi - https://fishesofaustralia.net.au.

characteristics and genetic similarities, demonstrating a coherent relationship between the two aspects. No evidence of taxonomic deviation was found in any of the species' group, indicating that the DNA barcoding approach accurately identified the analysed species.

**Table 2. Summary of the genetic divergence based on (K2P percentage) at each taxonomic level.**

| Label | n | Taxa | Comparisons | Min Dist (%) | Mean Dist (%) | Max Dist (%) | SE Dist (%) |
|---|---|---|---|---|---|---|---|
| Within Species | 352 | 101 | 504 | 0.00 | 0.21 | 1.51 | 0.00 |
| Within Genus | 166 | 25 | 399 | 3.64 | 12.53 | 23.43 | 0.01 |
| Within Family | 246 | 19 | 2002 | 8.27 | 17.67 | 25.52 | 0.00 |

### 3.2 Genetic distances and barcoding gap

The Kimura-2-parameter model (K2P) was used to calculate the genetic distance within and between species and the presence of barcode gap. The overall K2P distance of COI sequences are shown in (Table 2). Intraspecific distances ranged from 0.00% to 1.51% (within Species), while interspecific distances ranged from 3.64% to 23.43% (within Genus), which is greater than the minimum intraspecific distance of 2%, the threshold for the identification of fish species by DNA barcoding [58, 59] (Table 2 and Fig 3A and 3B).

This distance increased with increasing taxonomic level, with distance within families varying from a minimum of 8.27 to a maximum of 25.52 (within Family). The genetic distance of all species did not exceed the 2%, with the highest intraspecific distance belonging to *Acanthurus mata* OTU-4, with a genetic distance of 1.51% and a barcode gap. The remaining species had distances less than 1%. The minimum nearest neighbor genetic distance was 3.64% from *Epinephelus coioides* OTU-42 to *Epinephelus malabaricus* OTU-41, and the maximum was detected between *Hippocampus kelloggi* OTU-52 and *Carcharhinus leucas* OTU-19, with a genetic distance of 22.08%. All details regarding the species distance are summarized on the (S3 Table).

### 3.3 Revealing hidden fish diversity

Four species were unexpectedly detected for the first time in this geographical area, whose presence was not expected, including: *Boleophthamus dussumieri*, *Hippocampus kelloggi*, *Lethrinus miniatus* and *Maculabatis gerrardi*. The specie *Boleophthalmus dussumieri* is known to be endemic from the Persian Gulf region down to the Indian region [2], and *Lethrinus miniatus* from the Western Pacific [60]. For *B. dussumieri*, our molecular analysis has shown that

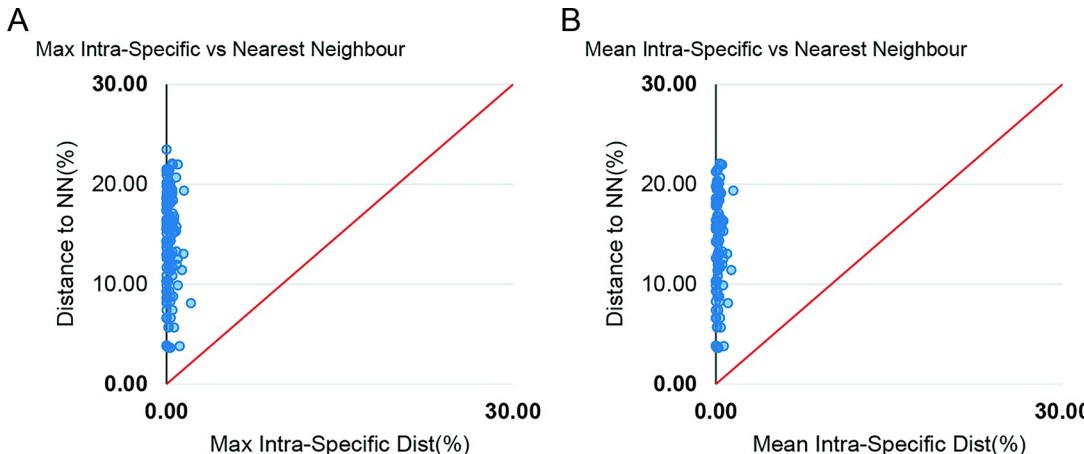

**Fig 3. Barcoding gap: Maximum intraspecific Kimura 2-parameter (K2P) distances compared with the minimum interspecific K2P distances recorded in fish species from the coast of Mozambique.** The graphs show the overlap of the maximum and mean intra-specific distances with the inter-specific (NN = nearest neighbour) distances.

there is more than one lineage that has not been discussed previously. On the other hand, the data also showed the presence of the Great Seahorse (*Hippocampus kellogi*) in the Central Province of Mozambique. This specie is listed as Vulnerable in the IUCN Red List, and its occurrence is known only from the Indo-Pacific region of Southeast Asia.

The other significant finding from our study was the presence of one of the most invasive fish species, the silver carp (*Hypophthalmichthys molitrix*), which is native to East Asia. This species is classified as "Near Threatened" and has spread globally, including in the Limpopo River in southern Mozambique. *Rhinoptera sp* was also recorded in this study. However, the occurrence of this species in Mozambique is not officially listed in the global fisheries database. Our molecular data also allowed us to detect for the first time, the presence of *Pangasius djambal* [61], which is known from Southeast Asia, *Dentex macrophthalmus*, which is distributed from the Mediterranean Sea to the eastern Atlantic [62], and *Fistularia petimba*, which is found in the western Atlantic [63].

### 3.4 Keystones species and conservation status

Sharks, stingrays, and seahorses are the most threatened species in the world due to their importance in commercial trade and use for various purposes, and therefore, the conservation and management of these species must be carefully ensured. In our study, we recorded fourteen endangered species (Table 1, Fig 2). Most of them were classified as elasmobranchs: *Acroteriobatus leucospilus*, Endangered (EN); *Carcharhinus leucas*, Vulnerable (VU); *Himantura uarnak*, Endangered (EN); *Loxodon macrorhinus*, Near Threatened (NT); *Maculabatis gerrardi*, Endangered (EN); *Pastinachus ater*, Vulnerable (VU); *Rhynchobatus djiddensis* and *Rhynchobatus australiae*, Critically Endangered (CR); *Sphyrna lewini*, Critically Endangered (CR); and one Syngnathiformes, *Hippocampus kellogi*, Vulnerable (VU).

Most of the teleost species identified in this study were classified as Least Concern in the IUCN Red List. However, some of them presented a certain level of threat: *Acanthopagrus vagus*, Vulnerable (VU); *Argyrosomus japonicus*, Endangered (EN), *Argyrosomus thorpei*, Endangered (EN), *Hypophthalmichthys molitrix*, Near Threatened (NT) and *Scomberomorus commerson*, Near Threatened (NT).

## 4. Discussion

This study has demonstrated the utility of DNA barcoding to supplement morphological identification. These results are particularly important because, while the DNA barcoding method has been applied globally to identify fish species, this study provides, for the first time, an overview of the marine fish diversity in the coastal region of Mozambique. This will make a significant contribution to our understanding of the understudied diversity in the Indo-West Pacific biogeographic region.

This study substantially improves our comprehension of fish species and families inhabiting the Mozambique Coast. The prominent occurrence of species within particular families is attributable to both the availability of specimens in the examined areas and the presence of economically important fish species. For instance, the Labridae family is especially abundant in southern Mozambique [64, 65]. In terms of family representation, this research ranks fourth, identifying 60 distinct families. This number is lower than those reported in previous studies, which recorded 90 and 94 families respectively, all using morphological identification methods [65]. Since morphological identification is prone to errors that can lead to misclassification of species, this study presents the most accurate and reliable assessment of fish diversity along Mozambique's coastline to date.

Accurate DNA barcoding species identification relies on their intra- and interspecific genetic distances. Consequently, closely related species may be readily detected in nearby geographic regions of occurrence. To be considered as species divergence, intraspecific genetic distance values must exceed 2%, which is the established threshold for species delimitation [41]. However, no specific threshold has been defined for interspecific or interfamily genetic distances. In the current study, the mean K2P genetic distances for the barcode region of the COI gene at the species, genera, and family levels are: 0.21±0.00%, 12.57±0.01%, and 17.29 ±0.00%, respectively (Table 2). The average of K2P genetic distances among species, genera, and families in this study was analogous to findings reported in other studies [31, 66, 67]. However, the minimum interspecific distance observed was substantially greater than that reported in [68–70] for fish in the China Sea, which can be ascribed to the increased taxonomic diversity of fish species in the present study.

In the present study, six species had low genetic distance from each other and clustered closely and two of them were not identified at specie level. The nearest neighbour of *Rhynchobatus australiae* is *R. djadensis* with 3.72%, *Oreochromis mossambicus* X *O. niloticus* with 3.68%, *Epinephelus malabaricus* X *E. coioides*, *Selar crumenophthalmus* X *Selar sp* and *Caesio caerulaurea* X *Caeseo sp* (Fig 2). These species, which we could not identify at species level, suggest that they belong to the same genus. Therefore, it is very likely that the sequence of these two morphospecies belongs to the same genus [71].

The barcode gap method was highly effective in distinguishing and delimiting all possible species. Despite the identification of 143 species represented by their respective operational taxonomic units (OTUs), some of these OTUs had remarkably low intraspecific distance and were treated as single species, yet were grouped into separate OTUs, such as *Molgarda sp1* (OTU-72), *Molgarda sp2* (OTU-73), and *Selar sp*. (OTU-138), and *Selar crumenophthalmus* (OTU-139) (S2 Table). This scenario arises when there is insufficient data in public databases to match the analysed specimens.

## 4.1 Revealing hidden fish diversity

The Indo-West Pacific region is renowned for its high diversity of fish species; however, there have been limited or inadequate studies conducted on the composition of fish fauna in the western Indian Ocean. This lack of information can result in an overestimation or underrepresentation of the biodiversity in this region. As an example, this study provides valuable data on the enigmatic presence of the amphibious fish species *B. dussumieri*, whose distribution was previously known to be restricted to the coast of Iran, across the Arabian Sea, and into the Indian region [72, 73].

This study also reports the presence of the stingray species *Maculabatis gerrardi* for the first time. This species is recognized as a complex of species within the Indo Pacific region, with three distinct lineages distributed along its geographic range [74]. Through the analysis of our COI gene sample and available COI gene sequences from GenBank and the BOLD System, our samples were found to cluster with *M. gerrardi* samples from Kwazulu-Natal (JF493650.1, JF493649.1) and Durban (JF493648.1) in South Africa, and the detection of an additional possible lineage in southern Africa suggests the need for further taxonomic examination of this group of stingrays.

Our data has demonstrated the occurrence of *Hippocampus kelloggi*, which significantly expands the known distribution range of the species, previously reported in the regions of China, India, Indonesia, Japan, Malaysia, Philippines, Thailand, Vietnam, and Tanzania [75]. The study confirms the presence of silver carp *Hypophthalmichthys molitrix* in the Limpopo estuary of southern Mozambique, which is widely regarded as one of the most introduced fish

species worldwide [76]. The exact circumstances surrounding its introduction to Mozambique for aquaculture purposes, remain unclear due to the inexistence in the list of produced specie in Aquaculture. It is possible that the presence of this species in the region may be attributed to flood runoff from South Africa [77].

The presence of *Lethrinus miniatus* in the western Indian Ocean is surprising, as it is generally found in the Western Pacific, the Ryukyu Islands, eastern Philippines, northern Australia, and New Caledonia [78]. Previous records of its occurrence outside of this range are misidentifications with another species of the same genus, *Lethrinus olivaceus*. However, our molecular data, when compared with studies conducted by [79–81], affirms the existence of *Lethrinus miniatus* in Mozambican waters.

Our study also documented the presence of *Fistularia petimba*. This species has been reported in several studies as spreading into new areas, including the archipelago of the Azores, Portugal [82], the Aegean Sea [83], the southern Iberian Peninsula [84], the Mediterranean Sea [85, 86], and the Syrian coast [87]. Notably, *Fistularia petimba* has also been recorded in the Gulf of Carpentaria in Australia [88]. Up until now, there have been no previous confirmed reports of its presence in the waters of Mozambique.

In summary, we provide strong evidence of the effectiveness of using DNA barcoding to accurately discriminate and identify many marine and coastal fish species examined thus far. The COI divergence patterns corresponded with morphologically recognized species; however, in some cases, the molecular data revealed previously undetected genetic divergence within a group and instances of low interspecific variation. The presence of cryptic taxa is relatively common among marine animals, emphasizing the need to consider the possibility of neglected diversity and the occurrence of species complexes. The more comprehensive the barcode library, the more useful it will be for the Barcode of Life Initiative [89]. This study represents a significant contribution towards consolidating DNA barcoding as a global system for identifying life forms and enhancing our understanding of the genetic diversity of Mozambican marine fish.

However, even among experienced taxonomists, consistent application of species names remains a challenge, especially when cryptic diversity is present. This is reflected in the conflicting names applied to specimens within the same BIN by different research groups. As the collection of global data increases, barcodes and BINs will play a vital role in integrating taxonomic feedback and significantly contributing to the standardization of names at the international level. This standardization is crucial for the sustainable management of the world's fisheries.

## Supporting information

**S1 Table. List of the identified species with detailed taxonomic information.** The list of species is from Eschmeyer's Catalog of Fishes and GBIF.
(XLSX)

**S2 Table. Cluster sequence result.** The distribution of sequence divergence at each taxonomic level based on the Pairwise Distance model.
(DOCX)

**S3 Table. Summary of the distance based on the K2P model.** The Distance Summary reports the sequence divergence between barcode sequences at the species, genus, and family level, and contrasts the distribution of within-species divergence to between-species divergence.
(PDF)

**S1 Fig. Phylogenetic tree (NJ) of all studied species.**
(EPS)

## Acknowledgments

We thank Manuel Carlos Lopes, Arnaldo Rumieque, Eusébio Fiquisse, Lucinda Cuamba, Afonso Muhala, Celia Cassamo, Osvaldo Muhala, Graça Malavi, Iolanda Lameira and Gertrudes Ussene for their contribution in acquiring and systematically identifying part of the samples.

## Author Contributions

**Conceptualization:** Valdemiro Muhala, Aurycéia Guimarães-Costa, Adam Rick Bessa-Silva, Iracilda Sampaio.

**Data curation:** Valdemiro Muhala, Luan Pinto Rabelo, Luciana Watanabe, Gisele Damasceno dos Santos.

**Formal analysis:** Valdemiro Muhala, Aurycéia Guimarães-Costa, Luan Pinto Rabelo, Luciana Watanabe, Gisele Damasceno dos Santos.

**Investigation:** Valdemiro Muhala, Isadola Eusébio Macate, Luísa Sambora.

**Methodology:** Valdemiro Muhala, Isadola Eusébio Macate, Luciana Watanabe, Luísa Sambora.

**Project administration:** Iracilda Sampaio.

**Resources:** Marcelo Vallinoto, Iracilda Sampaio.

**Software:** Valdemiro Muhala, Luan Pinto Rabelo.

**Supervision:** Aurycéia Guimarães-Costa, Adam Rick Bessa-Silva, Marcelo Vallinoto, Iracilda Sampaio.

**Validation:** Valdemiro Muhala, Marcelo Vallinoto, Iracilda Sampaio.

**Visualization:** Iracilda Sampaio.

**Writing – original draft:** Valdemiro Muhala.

**Writing – review & editing:** Valdemiro Muhala, Aurycéia Guimarães-Costa, Adam Rick Bessa-Silva.

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
