## [Decision Letter · Decision Letter 0]

30 Jun 2023

PONE-D-23-13181DNA barcoding for the assessment of marine and coastal Fish Diversity from the Coast of MozambiquePLOS ONE

Dear Dr. Muhala, 

Thank you for submitting your manuscript to PLOS ONE. After careful consideration, we feel that it has merit but does not fully meet PLOS ONE’s publication criteria as it currently stands. Therefore, we invite you to submit a revised version of the manuscript that addresses the points raised during the review process.

We look forward to receiving your revised manuscript.

Kind regards,

Feng ZHANG, Ph.D.

Academic Editor

PLOS ONE

Journal Requirements:

"This research was financed by Conselho Nacional de Desenvolvimento Científico e Tecnológico (CNPq) through a research project 407536/2021-3, 309916/2021-6 and the APC was funded by Pro-Reitoria de Pesquisa e Pós-Graduação of the Universidade Federal do Pará."

4. Thank you for stating the following in your Competing Interests section: "The authors declare no conflict of interest." 

7. We note that you have referenced (ie. Robertson, W. D., et al. [18]) which has currently not yet been accepted for publication. Please remove this from your References and amend this to state in the body of your manuscript: (ie “Robertson, W. D. et al. [Unpublished]”) as detailed online in our guide for authors

8. We note that Figure 1 in your submission contain map images which may be copyrighted. All PLOS content is published under the Creative Commons Attribution License (CC BY 4.0), which means that the manuscript, images, and Supporting Information files will be freely available online, and any third party is permitted to access, download, copy, distribute, and use these materials in any way, even commercially, with proper attribution. For these reasons, we cannot publish previously copyrighted maps or satellite images created using proprietary data, such as Google software (Google Maps, Street View, and Earth). For more information, see our copyright guidelines: http://journals.plos.org/plosone/s/licenses-and-copyright.

(1) You may seek permission from the original copyright holder of Figure 1 to publish the content specifically under the CC BY 4.0 license.  

9. We note that Figure 1 in your submission contain copyrighted images. All PLOS content is published under the Creative Commons Attribution License (CC BY 4.0), which means that the manuscript, images, and Supporting Information files will be freely available online, and any third party is permitted to access, download, copy, distribute, and use these materials in any way, even commercially, with proper attribution. For more information, see our copyright guidelines: http://journals.plos.org/plosone/s/licenses-and-copyright.

(1) You may seek permission from the original copyright holder of Figure 1 to publish the content specifically under the CC BY 4.0 license. 

(2) If you are unable to obtain permission from the original copyright holder to publish these figures under the CC BY 4.0 license or if the copyright holder’s requirements are incompatible with the CC BY 4.0 license, please either i) remove the figure or ii) supply a replacement figure that complies with the CC BY 4.0 license. Please check copyright information on all replacement figures and update the figure caption with source information. 

If applicable, please specify in the figure caption text when a figure is similar but not identical to the original image and is therefore for illustrative purposes only.

Reviewers' comments:

Reviewer's Responses to Questions

**Comments to the Author**

1. Is the manuscript technically sound, and do the data support the conclusions?

Reviewer #1: Yes

2. Has the statistical analysis been performed appropriately and rigorously? 

Reviewer #1: I Don't Know

3. Have the authors made all data underlying the findings in their manuscript fully available?

Reviewer #1: No

4. Is the manuscript presented in an intelligible fashion and written in standard English?

Reviewer #1: Yes

5. Review Comments to the Author

Reviewer #1: This paper can be acceptable by revising and addressing the following issues. Need to access the sequences conducted in this study for species identity check. Without accessing the species, I will not confirm the species’ accuracy. I want to see the sequences to comment on phylogenetic analysis and species identity.

1. In material and methods, please mention how you can match the valid and accepted species name and families. Authors should follow Eschmeyer's Catalog of Fishes for valid and accepted order, family, and scientific name.

2. The author submits the sequence data into the BOLD database. But did not mention the BIN number in the paper. Please include the BIN number in Table 1. Please be clear about the data availability specifically the sequences, so that reader can find your sequences. I want to see the sequences to comment on phylogenetic analysis and species identity. I encourage the author to submit sequences to the NCBI Gene Bank.

3. Author uses the GBIF database for taxonomic clarification. In the GBIF, you did not find the recent name for all species. So, you should follow Eschmeyer's Catalog of Fishes for valid and accepted order, family, and scientific name

4. In Table 1, Omit the genus column. Please write the species name in the elaborate species column. Use the border in column and row for family range for easy understanding to the reader. Follow the alphabetic order for species of the same family. Please include the Gene Bank accession number in the table for every species. Check the valid and accepted order, family, and scientific name by following Eschmeyer's Catalog of Fishes.

5. In the part of Revealing hidden fish diversity, Write the species name in alphabetic order. such as Boleophthamus dussumeri, Hippocampus kelloggi, Lethrinus miniatus and Maculabatis

6. Please change Figure 1. Please include a clear picture of the phylogeny to understand the phylogenetic position of species.

7. In the supporting information of Table S1, please follow Eschmeyer's Catalog of Fishes rather than following IUCN for the most updated synonym.

6. PLOS authors have the option to publish the peer review history of their article (what does this mean?). If published, this will include your full peer review and any attached files.

Reviewer #1: **Yes: **Md Jayedul Jslam

---

## [Author Response · Author response to Decision Letter 0]

30 Aug 2023

Journal Requirements:

R: We appreciated your thorough observations. All necessary adjustments have been made. Thank you for your valuable input.

R: Thank you for pointing that out. We will ensure that the correct grant numbers are provided in the 'Funding Information' section when we resubmit the article. Your feedback is appreciated. 

3. Thank you for stating the following financial disclosure: "This research was financed by Conselho Nacional de Desenvolvimento Científico e Tecnológico (CNPq) through a research project 407536/2021-3, 309916/2021-6 and the APC was funded by Pro-Reitoria de Pesquisa e Pós-Graduação of the Universidade Federal do Pará." Please state what role the funders took in the study. If the funders had no role, please state: "The funders had no role in study design, data collection and analysis, decision to publish, or preparation of the manuscript."

If this statement is not correct you must amend it as needed. Please include this amended Role of Funder statement in your cover letter; we will change the online submission form on your behalf.

R: Thank you for your comment. The funders had no role in the study design, data collection and analysis, decision to publish, or preparation of the manuscript. 

4. Thank you for stating the following in your Competing Interests section: "The authors declare no conflict of interest." Please complete your Competing Interests on the online submission form to state any Competing Interests. If you have no competing interests, please state "The authors have declared that no competing interests exist.", as detailed online in our guide for authors at http://journals.plos.org/plosone/s/submit-now This information should be included in your cover letter; we will change the online submission form on your behalf.

R: Thank you. We will update the Competing Interests section as per your instructions in the online submission form and include the appropriate statement in our cover letter.

R: We appreciate your comments. We want to assure you that we are fully committed to providing the necessary accession numbers or DOIs for our data upon manuscript acceptance. All sequences have been submitted to GenBank and we have chosen to keep the sequences restricted until this manuscript is accepted if that occurs. However, the data has already been submitted to Dryad and can be accessed from this link https://datadryad.org/stash/share/KRaWuznrD3B04KGLVyuzOqNUpal46PbrNNiBmMvz8K4

R: Thank you. We have included the full ethics statement in the 'Methods' section, providing all the necessary details as requested (Lines 157-163).

7. We note that you have referenced (ie. Robertson, W. D., et al. [18]) which has currently not yet been accepted for publication. Please remove this from your References and amend this to state in the body of your manuscript: (ie “Robertson, W. D. et al. [Unpublished]”) as detailed online in our guide for authors http://journals.plos.org/plosone/s/submission-guidelines#loc-reference-style.

R: You have been thanked for your guidance regarding the reference to the work of Robertson, W. D., et al. The suggested changes have been made, replacing the citation in the body of the text to "Robertson, W. D. et al. [Unpublished]". The reference list will be updated as directed. (Lines 62-63)

8. We note that Figure 1 in your submission contain map images which may be copyrighted. All PLOS content is published under the Creative Commons Attribution License (CC BY 4.0), which means that the manuscript, images, and Supporting Information files will be freely available online, and any third party is permitted to access, download, copy, distribute, and use these materials in any way, even commercially, with proper attribution. For these reasons, we cannot publish previously copyrighted maps or satellite images created using proprietary data, such as Google software (Google Maps, Street View, and Earth). For more information, see our copyright guidelines: http://journals.plos.org/plosone/s/licenses-and-copyright. We require you to either (1) present written permission from the copyright holder to publish these figures specifically under the CC BY 4.0 license, or (2) remove the figures from your submission:

(1) You may seek permission from the original copyright holder of Figure 1 to publish the content specifically under the CC BY 4.0 license. 

The following resources for replacing copyrighted map figures may be helpful: USGS National Map Viewer (public domain): http://viewer.nationalmap.gov/viewer/

R: We thank the careful review of our article and the valuable comments provided by the editorial team of PLOS. Regarding the concerns raised about the presence of map images that may be protected by copyright, we would like to clarify some important matters.

Firstly, we want to ensure that All shapes used in our article are from open access sources and freely available, obtained from the Natural Earth website (https://www.naturalearthdata.com/ ). Furthermore, it is essential to mention that the maps were created using the QGIS software (https://qgis.org/en/site/ ), which is a widely used open-source tool for geospatial analysis. We have included captions in Figure 1 that provide information about the origin of the shapes and the software used.

Regarding the photographs taken at the collection sites, we wish to clarify that all of them were taken by the first author of this manuscript (Valdemiro Muhala), and we are willing to provide any necessary evidence to confirm this information.

We are committed to ensuring the integrity of our work and to strictly adhere to the guidelines. We appreciate the guidance provided by the PLOS editorial team and hope that these actions will be sufficient to address the issue.

If there is any specific further guidance or if it is necessary to provide more information, please do not hesitate to inform us. We are eager to contribute to the advancement of knowledge and are grateful for the opportunity to resubmit our work for consideration by PLOS.

9. We note that Figure 1 in your submission contain copyrighted images. All PLOS content is published under the Creative Commons Attribution License (CC BY 4.0), which means that the manuscript, images, and Supporting Information files will be freely available online, and any third party is permitted to access, download, copy, distribute, and use these materials in any way, even commercially, with proper attribution. For more information, see our copyright guidelines: http://journals.plos.org/plosone/s/licenses-and-copyright.

(1) You may seek permission from the original copyright holder of Figure 1 to publish the content specifically under the CC BY 4.0 license. 

(2) If you are unable to obtain permission from the original copyright holder to publish these figures under the CC BY 4.0 license or if the copyright holder’s requirements are incompatible with the CC BY 4.0 license, please either i) remove the figure or ii) supply a replacement figure that complies with the CC BY 4.0 license. Please check copyright information on all replacement figures and update the figure caption with source information. 

If applicable, please specify in the figure caption text when a figure is similar but not identical to the original image and is therefore for illustrative purposes only.

R: We believe that these comments are the same as those made above. We once again appreciate the significant contribution. We will repeat here the response provided above.

We thank the careful review of our article and the valuable comments provided by the editorial team of PLOS. Regarding the concerns raised about the presence of map images that may be protected by copyright, we would like to clarify some important matters.

Firstly, we want to ensure that all shapes used in our article are from open access sources and freely available, obtained from the Natural Earth website (https://www.naturalearthdata.com/ ). Furthermore, it is essential to mention that the maps were created using the QGIS software ( https://qgis.org/en/site/), which is a widely used open-source tool for geospatial analysis. We have included captions in Figure 1 that provide information about the origin of the shapes and the software used.

Regarding the photographs taken at the collection sites, we wish to clarify that all of them were taken by the first author of this manuscript (Valdemiro Muhala), and we are willing to provide any necessary evidence to confirm this information.

We are committed to ensuring the integrity of our work and to strictly adhere to the guidelines. We appreciate the guidance provided by the PLOS editorial team and hope that these actions will be sufficient to address the issue.

If there is any specific further guidance or if it is necessary to provide more information, please do not hesitate to inform us. We are eager to contribute to the advancement of knowledge and are grateful for the opportunity to resubmit our work for consideration by PLOS.

10. Please review your reference list to ensure that it is complete and correct. If you have cited papers that have been retracted, please include the rationale for doing so in the manuscript text or remove these references and replace them with relevant current references. Any changes to the reference list should be mentioned in the rebuttal letter that accompanies your revised manuscript. If you need to cite a retracted article, indicate the article’s retracted status in the References list and also include a citation and full reference for the retraction notice.

R: We appreciate the suggestions. All references have been reviewed. 

Reviewers' comments:

Reviewer's Responses to Questions

Comments to the Author

1. Is the manuscript technically sound, and do the data support the conclusions? The manuscript must describe a technically sound piece of scientific research with data that supports the conclusions. Experiments must have been conducted rigorously, with appropriate controls, replication, and sample sizes. The conclusions must be drawn appropriately based on the data presented.

Reviewer #1: Yes

2. Has the statistical analysis been performed appropriately and rigorously?

Reviewer #1: I Don't Know

3. Have the authors made all data underlying the findings in their manuscript fully available?

The PLOS Data policy requires authors to make all data underlying the findings described in their manuscript fully available without restriction, with rare exception (please refer to the Data Availability Statement in the manuscript PDF file). The data should be provided as part of the manuscript or its supporting information or deposited to a public repository. For example, in addition to summary statistics, the data points behind means, medians and variance measures should be available. If there are restrictions on publicly sharing data e.g., participant privacy or use of data from a third party—those must be specified.

Reviewer #1: No

4. Is the manuscript presented in an intelligible fashion and written in standard English? PLOS ONE does not copyedit accepted manuscripts, so the language in submitted articles must be clear, correct, and unambiguous. Any typographical or grammatical errors should be corrected at revision, so please note any specific errors here.

Reviewer #1: Yes

5. Review Comments to the Author

Please use the space provided to explain your answers to the questions above. You may also include additional comments for the author, including concerns about dual publication, research ethics, or publication ethics. (Please upload your review as an attachment if it exceeds 20,000 characters).

Reviewer #1: This paper can be acceptable by revising and addressing the following issues. Need to access the sequences conducted in this study for species identity check. Without accessing the species, I will not confirm the species’ accuracy. I want to see the sequences to comment on phylogenetic analysis and species identity.

R: Thank you for your concern regarding our manuscript. The data has been submitted to the GenBank, and now, it is still under restricted access. However, we will deposit the database in the Dryad repository for your evaluation ( https://datadryad.org/stash/share/KRaWuznrD3B04KGLVyuzOqNUpal46PbrNNiBmMvz8K4 )

1. In material and methods, please mention how you can match the valid and accepted species name and families. Authors should follow Eschmeyer's Catalog of Fishes for valid and accepted order, family, and scientific name.

R: Thank you for your suggestion. We will revise our methodology to clearly include the use of Eschmeyer's Catalog of Fishes for species and family names verification (Lines 222-224)

2. The author submits the sequence data into the BOLD database. But did not mention the BIN number in the paper. Please include the BIN number in Table 1. Please be clear about the data availability specifically the sequences, so that reader can find your sequences. I want to see the sequences to comment on phylogenetic analysis and species identity. I encourage the author to submit sequences to the NCBI Gene Bank.

R: Thank you for your comments. As per your request, we have included the BINs in Table 1. As mentioned earlier, the sequences have already been submitted and are currently under restricted access. However, we want to emphasize that we will also submit the database used for your evaluation. It can be accessed at this link: https://datadryad.org/stash/share/KRaWuznrD3B04KGLVyuzOqNUpal46PbrNNiBmMvz8K4.

3. Author uses the GBIF database for taxonomic clarification. In the GBIF, you did not find the recent name for all species. So, you should follow Eschmeyer's Catalog of Fishes for valid and accepted order, family, and scientific name.

R: Thank you for the observation. We reviewed the information and considered your suggestion.

4. In Table 1, Omit the genus column. Please write the species name in the elaborate species column. Use the border in column and row for family range for easy understanding to the reader. Follow the alphabetic order for species of the same family. Please include the Gene Bank accession number in the table for every species. Check the valid and accepted order, family, and scientific name by following Eschmeyer's Catalog of Fishes.

R: Thank you for the suggestions. We have made all the suggested changes.

5. In the part of Revealing hidden fish diversity, Write the species name in alphabetic order. such as Boleophthamus dussumeri, Hippocampus kelloggi, Lethrinus miniatus and Maculabatis gerrardi

R: Thank you for your feedback. We arranged the species names alphabetically as suggested.

6. Please change Figure 1. Please include a clear picture of the phylogeny to understand the phylogenetic position of species.

R: Thank you for your feedback. If I understand correctly, you are referring to Image 2. We will revise Figure 2 and include a clear picture of the phylogeny to help you better understand the content.

7. In the supporting information of Table S1, please follow Eschmeyer's Catalog of Fishes rather than following IUCN for the most updated synonym.

R: Thank you for your comment. We will update the supporting information of Table S1 following Eschmeyer's Catalog of Fishes for the most updated synonym.

---

## [Editor Report · Decision Letter 1]

11 Oct 2023

DNA barcoding for the assessment of marine and coastal Fish Diversity from the Coast of Mozambique

PONE-D-23-13181R1

Dear Dr. Valdemiro Muhala,

We’re pleased to inform you that your manuscript has been judged scientifically suitable for publication and will be formally accepted for publication once it meets all outstanding technical requirements.

Kind regards,

Feng ZHANG, Ph.D.

Academic Editor

PLOS ONE
---

## [Editor Report · Acceptance letter]

16 Oct 2023

PONE-D-23-13181R1 

DNA barcoding for the assessment of marine and coastal Fish Diversity from the Coast of Mozambique 

Dear Dr. Muhala:

I'm pleased to inform you that your manuscript has been deemed suitable for publication in PLOS ONE. Congratulations! Your manuscript is now with our production department. 

Kind regards, 

on behalf of

Dr. Feng ZHANG 

Academic Editor

PLOS ONE